# Zika Virus Infects Human Placental Mast Cells and the HMC-1 Cell Line, and Triggers Degranulation, Cytokine Release and Ultrastructural Changes

**DOI:** 10.3390/cells9040975

**Published:** 2020-04-16

**Authors:** Kíssila Rabelo, Antônio José da Silva Gonçalves, Luiz José de Souza, Anna Paula Sales, Sheila Maria Barbosa de Lima, Gisela Freitas Trindade, Bianca Torres Ciambarella, Natália Recardo Amorim Tasmo, Bruno Lourenço Diaz, Jorge José de Carvalho, Márcia Pereira de Oliveira Duarte, Marciano Viana Paes

**Affiliations:** 1Laboratório de Ultraestrutura e Biologia Tecidual, Universidade do Estado do Rio de Janeiro, Rio de Janeiro 20551-030, Brazil; bia.btc2@gmail.com (B.T.C.); jjcarv@gmail.com (J.J.d.C.); 2Laboratório Interdisciplinar de Pesquisas Médicas, Instituto Oswaldo Cruz, Rio de Janeiro 21040-900, Brazil; ajsgoncalves@gmail.com; 3Faculdade de Medicina de Campos, Campos dos Goytacazes, Rio de Janeiro 28035-581, Brazil; luizjosedes@gmail.com (L.J.d.S.); ap.sss@hotmail.com (A.P.S.); 4Laboratório de Tecnologia Virológica, Biomanguinhos, Rio de Janeiro 21040-900, Brazil; smaria@bio.fiocruz.br (S.M.B.d.L.); gisela.freitas@bio.fiocruz.br (G.F.T.); 5Laboratório de Inflamação, Instituto de Biofísica Carlos Chagas Filho, Universidade Federal do Rio de Janeiro, Rio de Janeiro 21941-170, Brazil; nataliaamorim1@hotmail.com (N.R.A.T.); bldiaz@biof.ufrj.br (B.L.D.)

**Keywords:** flavivirus, immune response, inflammatory mediator

## Abstract

Zika virus (ZIKV) is an emergent arthropod-borne virus whose outbreak in Brazil has brought major public health problems. Infected individuals have different symptoms, including rash and pruritus, which can be relieved by the administration of antiallergics. In the case of pregnant women, ZIKV can cross the placenta and infect the fetus leading to congenital defects. We have identified that mast cells in the placentae of patients who had Zika during pregnancy can be infected. This led to our investigation on the possible role of mast cells during a ZIKV infection, using the HMC-1 cell line. We analyzed their permissiveness to infection, release of mediators and ultrastructural changes. Flow cytometry detection of ZIKV-NS1 expression 24 h post infection in 45.3% of cells showed that HMC-1 cells are permissive to ZIKV infection. Following infection, β-hexosaminidase was measured in the supernatant of the cells with a notable release at 30 min. In addition, an increase in TNF-α, IL-6, IL-10 and VEGF levels were measured at 6 h and 24 h post infection. Lastly, different intracellular changes were observed in an ultrastructural analysis of infected cells. Our findings suggest that mast cells may represent an important source of mediators that can activate other immune cell types during a ZIKV infection, which has the potential to be a major contributor in the spread of the virus in cases of vertical transmission.

## 1. Introduction

Zika fever is an important *Arbovirus*-caused disease that has surfaced in numerous countries in Asia, Africa and America [1]. The etiological agent of this disease, Zika virus (ZIKV), was initially isolated in 1947 from the blood of sentinel *Rhesus* monkeys during a study on yellow fever transmission in the Zika forest of Uganda, which gave rise to its name [2,3]. Transmission of the ZIKV is primarily through bites of infected *Aedes* mosquitos, with the most common vectors being *Aedes aegypti* and *Aedes albopictus,* but it can also happen by vertical transmission [4,5]. As a result of vertical transmission, there were alarming cases of Congenital Zika Syndrome, as the virus could cause damage to the placenta, infect placental cells and reach the fetus [6]. A ZIKV particle has a diameter of 25–30 nm and is a member of the *Flaviviridae* family that shares many similarities with other more widely known related viruses such as dengue, West Nile, Japanese encephalitis and yellow fever [4,7]. It has a single-stranded RNA genome with a positive polarity of 11 Kb and encodes a polyprotein precursor that is processed into the structural proteins such as capsid (C), pre-membrane (prM) and envelope (E) along with seven non-structural proteins (NS1, NS2A, NS2B, NS3, NS4A, NS4B and NS5) [8,9].

Mast cells are resident immunological cells found abundantly in tissues such as skin, endometrium and placenta that have prominent roles in immunologic reactions [10,11,12,13]. Their presence and prevalence in these tissues, along with their proximity to blood vessels, predispose these cells to be among the first immune cells that can be infected by ZIKV after a mosquito bite penetrates the skin. As some of the most frequent symptoms of zika are rash and pruritus, which are relieved by the administration of antiallergic drugs (anti-histamines), this has led us to believe that mast cells can play a role, although not yet elucidated, in the pathogenesis of the disease [14,15,16]. We hypothesize that it may be one of the cells involved in placental infections, which can directly contribute to vertical transmission.

Although there are no studies in the literature that have investigated the involvement of mast cells in a ZIKV infection to date, mast cells have a proven role in infections by dengue, another *Arbovirus*. Several products originating from mast cells are found at high levels in patients infected by dengue, especially those with plasma leakage [17,18]. While mast cells are permissive to dengue infection, it is most probable that they display a low level of the specific receptors required since the quantity of virus necessary to successfully infect this cell type is always higher than is needed for macrophages and dendritic cells [17,19,20].

HMC-1 cells are a lineage of human mast cells that characteristically express the cytokine receptor c-Kit abundantly and release different cytokines after degranulation stimuli. This cell line possesses the features necessary to serve as an in vitro model for the development of studies on mast cells [21]. HMC-1 has been widely used in studies on degranulation studies, endothelial activation and its interaction with other arboviruses [22,23,24].

Here, we present our observations on the presence of mast cells in ZIKV-infected human placentae and observed viral replication in these cells. Additionally, we investigated the potential for ZIKV to infect HMC-1 cells as a model system for mast cells and quantified the percentage of infected cells in different MOIs. We further studied the degranulation of these cells after contact/infection with ZIKV by measuring β-hexosaminidase release as well as the expression profiles of TNF-α (tumor necrosis factor-α), IL-6 (interleukin-6), IL-10 (interleukin-10) and VEGF (vascular endothelial growth factor). As a final point, we evaluated the effects of ZIKV infection on the ultrastructure of HMC-1 cells. Together, the findings validate a critical and, to our knowledge, previously unrecognized role for mast cells in the infection and propagation of ZIKV in humans.

## 2. Materials and Methods

### 2.1. Placentae Collection, Patient Clinical History and Ethical Approval

At delivery, samples from the placentae were collected and fixed in 10% formaldehyde. Samples were collected at the Hospital Plantadores de Cana, Campos dos Goytacazes, RJ, Brazil. As a control, a sample of a full-term placenta from a healthy donor was included.

Case 1: A 23-year-old patient. Symptoms: fever, arthralgia, exanthema and pruritus in the third trimester of gestation. At 38 weeks of gestation, her baby girl was born by cesarean delivery, with 37 cm of cephalic circumference. The mother’s IgM serology was positive for Zika. The test for dengue NS1 was negative.

Case 2: A 34-year-old patient. Symptoms: exanthema and pruritus in the third trimester of gestation. Her baby girl was born at term, by cesarean delivery, with 38 weeks of gestation. She presented with a normal 34 cm of cephalic circumference. The mother’s IgM serology was positive for Zika.

Patient recruitment and the procedures performed were pre-approved by the Ethics Committee of the Oswaldo Cruz Foundation/FIOCRUZ (CAEE: 65924217.4.0000.5248) and by the Ethics Committee of Faculty of Campos Medicine/Benedito Pereira Nunes Foundation (CAEE: 65924217.4.3001.5244). The patients were fully informed of the research plans and provided written consent to participate, which included permission to publish all data without identifying information.

### 2.2. Histopathology and Histological Detection of Mast Cells in ZIKV Infected Placentae

All histological processing of the sample was performed as described previously by our group [25]. The histopathological analysis was performed on the images observed and captured by hematoxylin and eosin (H&E) staining. The staining used to highlight the mast cells was Toluidine Blue 1%. Stained specimens were visualized by light microscopy (Olympus, Tokyo, Japan), and digital images were obtained using Image-Pro Plus software version 7.0.

### 2.3. Immunofluorescence Assay

Immunofluorescence was performed as described in Rabelo et al., 2017 [25]. Antibodies were used at a dilution of 1:200 for a mouse monoclonal anti-Zika NS1 IgG (Arigo Biolaboratories, Taiwan, Republic of China), and a rabbit polyclonal antihuman c-Kit IgG (Santa Cruz, Texas, USA). After staining with primary antibodies, sections were incubated with an Alexa 488-conjugated rabbit anti-mouse IgG, Alexa 555-conjugated goat anti-rabbit IgG, or Alexa 555-conjugated goat anti-mouse IgG (ThermoFisher, Waltham, MA, USA). Slides were visualized by fluorescence microscopy (Olympus, Tokyo, Japan), and digital images were obtained using Image-Pro Plus software version 7.0.

### 2.4. Immunohistochemistry

The protocol for immunohistochemistry was described previously by our group [25]. Briefly, the slides were incubated overnight at 4 °C with a 1:200 dilution of the mouse monoclonal antibody IgG antibody against Zika NS1 (Arigo Biolaboratories, Taiwan, Republic of China). Then, sections were maintained with a rabbit anti-mouse IgG conjugated to horseradish peroxidase (Spring Bioscience Corporation, CA, USA) for 40 min at room temperature. We visualized the sections by light microscopy (Olympus, Tokyo, Japan), and digital images were obtained using Image-Pro Plus software version 7.0.

### 2.5. Cell Line

The HMC-1 cell line was kindly provided by Dr. Joseph H. Butterfield (Mayo Clinic, Rochester, NY, USA) and cultured in Iscove’s Modified Dulbecco’s Medium (IMDM- Thermo Fisher, Waltham, MA, USA) supplemented with 10% fetal bovine serum (FBS, Cultilab, Campinas, SP, Brazil), 40 U/mL penicillin/streptomycin (Sigma, St. Louis, MS, USA) and 1.2 mM α-thioglycerol (Sigma, St. Louis, MS, USA). Cells were maintained at 37 °C in a humidified incubator at 5% CO_2_. Culture media was exchanged every 3–4 days with splitting of cultures at a confluency of 80–90%.

### 2.6. ZIKV Viral Stock Productuion

A primary clinical virus specimen was isolated from a serum sample of a patient from Paraiba. The virus was propagated in a culture of C6/36 *Ae. albopictus* mosquito cells and harvested virus was tittered by the infection of Vero cells (CCL-81) followed by RT-PCRq, which determined a titer of 5.8 × 10^6^ PFU/mL. Copy numbers were assessed by using a standard curve in the RT-PCRq reaction containing 1 × 10^8^ copies/reaction. The oligonucleotide set utilized targeted the intergenic region of the Membrane/Envelope as described by Lanciotti, 2008 [26] (Table 1).

### 2.7. ZIKV Infections

Infections were performed by varying the multiplicity of infection (MOI) at 0.1, 0.2 and 1.0. ZIKV viral particles per host cell. Virus was added to cell culture and incubated for 1 h at 37 °C prior to removal of unattached viral particles and a further incubation of 6 h or 24 h. For a 30 min time point, virus was incubated with cells for 30 min before rinsing and preparation of flow cytometry. As a negative control, cells were incubated in the same conditions with a mock viral stock consisting of a supernatant of non-infected Vero cells.

### 2.8. Flow Cytometry Analysis

The expression of NS1 protein in infected HMC-1 cells was analyzed by flow cytometry. Cells were collected by centrifugation, and suspended in PBS for 30 min, 6 h or 24 h after infection with different MOIs. Approximately 10^6^ cells/well were fixed in 4% formaldehyde for 25 min and permeabilized with 0.05% saponin for 30 min. Next, cells were incubated with a 1:1000 dilution of the mouse monoclonal IgG antibody against ZIKV non-structural protein NS1 (Arigo Biolaboratories, Taiwan, Republic of China) for 1 h at 37 °C before being washed with PBS. This was followed by an incubation with a 1:200 dilution of an Alexa 488-conjugated anti-mouse (Thermo Fisher, Waltham, MA, USA) for 30 min. After washing with PBS, cells were suspended in PBS and applied to a flow cytometer (Facs Calibur; BD Biosciences, San Jose, CA, USA) to measure fluorescence, which was analyzed offline with Summit 6.1 software.

### 2.9. Measurement of Mast Cell Degranulation

Mast cell degranulation was evaluated by measuring the activity of the granule-stored enzyme- β-hexosaminidase that was secreted into the extracellular medium. Cells were infected with MOI 0.1, 0.2 or 1 in 6-well plates (1 × 10^6^/well) for 30 min. Aliquots of the supernatant (15 µl) were transferred to 96-well plates and incubated with 60 μL of substrate (1 mM p-nitrophenyl-N-acetyl-b-D-glucosaminide) in 0.05 M sodium citrate (pH 4.5) for 60 min at 37 °C. In addition, we used 60 μL of substrate solution (1 mM p-nitrophenyl-N-acetyl-β-D-glucosaminide (Sigma, St. Louis, MS, USA) in 100 mM sodium citrate, pH 4.5) and incubated for 60 min at 37 °C. Reactions were stopped by adding 150 μL of 0.1 M Na_2_CO_3_-NaHCO_3_ buffer (pH 10). Enzyme activity was measured as the absorbance at 405 nm. Total β-hexosaminidase activity was determined by releasing all enzyme through lysis with 0.1% Triton X-100 and measuring activity from a 15 µl aliquot. As a positive control for degranulation, we used 20 µg/mL of 48/80 compound (Sigma, St. Louis, MS, USA). The results are presented as the percentage of total β-hexosaminidase content of the cells.

### 2.10. ELISA Assays

The quantity of cytokines and factors released from mast cells by infection with ZIKV was measured by ELISA. Supernatants from HMC-1 cells infected at a MOI of 1 for 30 min, 6 h or 24 h were evaluated for IL-6 (900-T16), IL-10 (900-K21), TNF-α (900-T25) and VEGF (900-K10) with commercial ELISA assay kits (Peprotech Inc. Rocky Hill, NJ, USA), according to the manufacturer’s instructions.

### 2.11. Transmission Electron Microscopy Procedure

HMC-1 cells were infected with ZIKV at a MOI of 1 for 30 min or 24 h and then fixed with 2.5% glutaraldehyde in 0.1 M sodium cacodylate buffer (pH 7.2). Cells were post-fixed with 1% buffered osmium tetroxide, dehydrated in an acetone series (30, 50, 70, 90, and 100%) and then embedded in EPON (Electron Microscopy Sciences, Hatfield, PA, USA) through polymerization at 60 °C for 3 days. Ultrathin sections (60–90 nm) were contrasted with uranyl acetate and lead citrate before visualization using a JEOL 1001 transmission electron microscope (Jeol Ltd., Tokyo, Japan).

### 2.12. Statistical Analysis

Data were analyzed in GraphPad Prism software v 6.0 (GraphPad Software, San Diego, CA, USA) using non-parametric statistical tests. Significant differences between the analyzed groups were determined using the Mann–Whitney test with a threshold of *p* < 0.05.

## 3. Results

### 3.1. Detection of Mast Cells, Histopathology and ZIKV Replication in Placental Infected Tissues

First, we evaluated the presence of mast cells in the placentae of ZIKV infected women during pregnancy in comparison to a non-infected control sample. To detect mast cells, we performed immunohistochemistry with a Toluidine Blue stain and identified these cells in placental sections of these patients by the prominent purple coloration (Figure 1A–C, arrows). Next, fluorescence microscopy images (Figure 1D–F) were used to identify cells that displayed both the mast cell marker c-Kit (red) and ZIKV NS1 protein (green). As expected, no evidence of ZIKV NS1 protein was observed in control placenta (Figure 1D). In constrast, dually labeled cells were readily observed in placenta from both ZIKV seropostive patients (Figure 1E,F), which suggested that these cells were infected and supported virus replication (Figure 1E,F). To examine the histopathological aspects, H&E stainging was used to identify maternal portions (basal decidua) and fetal portions (chorionic villi), which were normal in the control placenta (Figure 1G). Within the placentae from the ZIKV infected patients, case 1 presented areas with immature chorionic villi, chronic villositis and chronic deciduitis with lymphocytes in chorionic villi and decidua (Figure 1H). The placenta from case 2 showed intervillitis with lymphocytes in the intevillous space and immature chorionic villi (Figure 1I). To extentd the search for cells supporting ZIKV replication, immunohistochemistry was used to provide broad staining of NS1 protein both in the maternal and fetal portions of the placentae. Again, the control, non-infected samples showed no reactivity against NS1. Within placentae from infected mothers, extensive reactivity was seen in not only immune cells, but also trophoblasts and decidual cells suggesting that they are also permissive to infection (Figure 1K,L).

### 3.2. Infection Rate of ZIKV at Different MOIs

After observing that placental mast cells were infected with ZIKV during a natural infection, the susceptibility to ZIKV entry and permissiveness to its replication was evaluated using the HMC-1 cell line under controlled conditions. Cells were exposed to three different MOIs (0.1, 0.2 and 1) of virus or an equal volume of mock as a control to determine conditions of infections. A mock viral stock was generated from supernatants of Vero cells that were not exposed to ZIKV as a control. The percentage of cells infected by ZIKV was determined by counting the number of cells displaying the fluorescent detection of NS1, a protein that is present only after viral replication, by flow cytometry. Cells were either incubated with virus or mock for 30 min and processed for analysis, or for 1 h with a subsequent incubation for 6 h or 24 h. NS1 was detected under all conditions (Figure 2A), even after 30 min, which suggests that ZIKV can rapidly enter cells and begin replication. Considering that the percentage of cells was nearly equivalent across the three MOIs at 30 min, the results further suggest that only a subset of cells were susceptible to rapid infection. By increasing the virus binding and entry time to 1 h, followed by a 6 h incubation, the percent of cells infected increased with a maximum percent observed with a MOI of 1. A slight increase in the percentage of cells was measured when the post-infection incubation increased to 24 h. Averaged histograms of the three conditions (Figure 2B–D) show a nearly equivalent low background from the mock and the highest levels of infection with a MOI of 1 in 6 h and 24 h, with a mean of 40.10 ± 4.81 and 45.30 ± 3.44% of infected cells in three independent experiments, respectively.

### 3.3. ZIKV Interaction Induces Degranulation

The results from the infection of mast cells by ZIKV suggested that the response of HMC–1 could be contributing to the observations. We chose to explore the activation and degranulation of mast cells by β-hexosaminidase, a resident enzyme released in response to degranulation. Initially, flow cytometry was used to analyze the percentage of cells that display degranulation following incubations of HMC-1 cells with ZIKV at different MOIs for different times. After a 30 min incubation, all three MOIs showed similar percentages (Figure 3A). The percentage of cells decreased following a 6 h incubation and returned to the levels of mock infections after a 24 h incubation suggesting that, at the later time points, the granulosome recuperated or the released enzyme lost activity.

To evaluate the early kinetics of mast cell activation, the amount of β-hexosaminidase, normalized to the total cellular β-hexosaminidase, was measured at 30 min for each of the MOIs (Figure 3B). Despite the β-hexosaminidase levels not reaching the percentage of the cells stimulated with the synthetic compound 48/80, the release was gradually increased according to the amount of viral particles, suggesting that the activation of these cells actually occurs due to adsorption of the virus to cell receptors.

### 3.4. ZIKV Led to Release of Cytokines and VEGF 

To analyze the release of the cytokines TNF- α, IL-6, and IL-10, along with VEGF, during infection with ZIKV, we performed ELISAs on the supernatant of mast cells activated with 30 min of contact with the virus as well as the extracellular levels produced by 1 h of virus presence and an incubation of 6 or 24 h. After the shortest interaction time, the levels of TNF-α, IL-6 and IL-10 increased greater in response to exposure to the control than with ZIKV (Figure 4). The levels of VEGF were nearly equal. The levels of the cytokines and VEGF in the supernatant were significantly greater after a hour incubation with ZIKV stocks with an additional 6 h incubation than the mock stocks. This difference grew with the increase in the secondary incubation time to 24 h although the absolute levels of these cytokines and VEGF were lower compared to 6 h. The observed release of TNF-α, IL-6, IL-10 and VEGF at 30 min suggested that they responded to a range of external stimuli. Meanwhile, the elevated levels of these mediators 6 h or 24 h after the infection with ZIKV infection suggests a stimulation in expression and release of these mediators after infection.

### 3.5. Ultrastructural Changes Caused by ZIKV Infection

To explore changes to aspects of the ultrastrutucture of mast cells in response to ZIKV infections, an infection with an MOI of 1 was used for the best conditions of infection as well as activation and degranulation of HMC-1. As a control for the analysis, the ultrastructure of cells incubated with the mock viral stock for 30 min was evaluted. Representative cells presented normal aspects for a mast cell in terms of the formation of the nucleus, and the volume of the mitochondria and normal endoplasmic reticulum with a high density of granules (Figure 5A–C). While cells incubated with the ZIKV for 30 min have a lower rate of infection, our previous data show they are at the optimal moment of adsorption and trigger degranulation. The ultrastructure of representative cells shows a decrease in cellular granules (Figure 5D–F), with no other major alterations. After 24 h incubation with the mock viral stock, we observed that the mastocytes continued to have a high density of granules, endoplasmic reticulum with closed cisterns, and mitochondria with some structural alterations, such as swollen and ruptured (Figure 5G–I). After the same period of incubation, the infected mast cells presented various organelle alterations observed as the formation of numerous vesicles, dilated endoplasmic reticulum cisterns, swollen mitochondria, ruptures in cellular membranes and, in some cells, the absence of a nucleus suggesting that a subset of cells may no longer be viable (Figure 5J–K). In several instances, the presence of viral-like particles were detected that match with the size of a ZIKV particle (Figure 5L).

## 4. Discussion

Mast cells have an important function in developing an inflammatory process and are present in a variety of tissues such as skin and mucous membranes that include the placenta. However, there have been no studies that have investigated the role of mast cells in ZIKV infection and its pathogenesis. Many studies have described a permissiveness and replication of ZIKV in different placental and immune cells [25,27,28]. These descriptions are extensive in relation to Hofbauer cells and deciduous macrophages [29,30,31]. Here, we report for the first time the detection of virus in vivo in mast cells present in placental tissue from two women seropositive for a ZIKV infection through the NS1 protein of Zika. Mast cells are resident cells in the endometrium and placenta, and it is believed that they can play multiple roles from implantation to placental immune response during pregnancy, including trophoblastic migration and angiogenesis [32,33]. The implications of ZIKV infections in placental mast cells could have some importance in understanding the inflammatory process and vertical transmission.

Based on the in situ results, we performed a series of experiments in vitro using the HMC-1 cell line as a model system for mast cells to unveil aspects of their interactions and reaction to infections by ZIKV. First, we observed by a flow cytometer analysis that the HMC-1 cells are able to support the entry of the virus as well as its rapid replication within 30 min. Replication was inferred by the detection of the NS1 protein of Zika, which is a non-structural protein that is not a constituent of the virus particle and is only present after its synthesis at the time of replication [9]. It is known that mast cells have the requisite receptors, such as FcɣR, HSP70 and others, that could mediate the entry of ZIKV, and also mediate the entry of other arboviruses like dengue, as well as being involved in the transduction signals for the degranulation cascade [21,34].

One of the most abundant proteases present in mast cell granules widely used to assess degranulation is β-hexosaminidase, a glycolytic enzyme that is released into the tissues and triggers typical reactions in allergy and inflammatory responses [35]. We used the quantification of β-hexosaminidase in cell supernatants as a measurement of mast cell degranulation as a result of incubations with ZIKV. We used the synthetic compound 48/80, which is a standard degranulator, to elicite β-hexosaminidase release by HMC-1 cells [36]. There was a significant increase in the release of β-hexosaminidase by the HMC-1 cell line after contact with ZIKV, which was only detected at 30 min, which leads us to believe that viral adsorption is a stimulus for degranulation. The time frame of 30 min is consistent with that of the adsorption and internalization of flavivirus particles, which occurs rapidly in 13 to 15 min as observed for the intracellular localization of DENV particles [37]. In MOI 1, β-hexosaminidase levels were near that of the positive control with 48/80. Degranulation, detected by the release of β-hexosaminidase, has been associated with the injection of DENV in other studies [19]. The cleavage of some substrates of this enzyme has been associated with NKT cell differentiation, and the high activity of β-hexosaminidase has already been observed in placental dysfunction [35,38].

In addition to the enzymes released during degranulation, mast cells are responsible for the production and release of different pro-inflammatory cytokines. We evaluated the production of TNF-α, IL-6 and IL-10 at different times from viral adsorption to 6 h and 24 h post infection. At the moment of initial contact of the mast cells with the mock or the virus, there was a release of these cytokines and VEGF, which is consistent with mast cells having internal stores that are primed for release in response to a stimulus. As the supernatant of Vero cells (mock) has a rich secretion of proteins, this stimulus appears to have been sufficient for the release within 30 min. However, at the end of other incubation times, the mock viral stock controls were associated with low secretion levels of these mediators, which contrasted with the ZIKV infected cells. There was a significant increase in levels of both cytokines at 6 h, which would be expected to generate an environment conducive to the recruitment and differentiation of other immune cells. TNF-α is produced for optimal defense against pathogens in inflammation resolution and orchestrates the tissue recruitment of immune cells and promotes tissue remodeling and destruction [39]. IL-6 is a cytokine with a crucial role in inflammation. It also leads to recruitment and differentiation of mast cells, as well as monocytes, CD4+ and CD8+ T cells, and B lymphocytes, and it stimulates the production of VEGF by fibroblasts. IL-6 expression affects the homeostatic processes that is related to tissue injury and activation of stress-related responses [40,41,42,43]. Despite being an anti-inflammatory cytokine, the expression of IL-10 was increased in ZIKV-infected HMC-1 cells, which corroborates what was observed in another study, in the serum of Zika positive patients [44]. Moreover, it is a cytokine normally produced by triggered mast cells, which leads to activation of other mast cells and is present in allergic responses [45,46]. In support of our findings, an increase in cytokines related to the inflammatory environment in placental ZIKV infection has already been observed in another study performed by our group, with an increase in TNF-α and the VEGFR-2 receptor [28]. TNF-α combined with VEGF were similarly related to vascular placental dysfunction, leading to plasma overflow and preeclampsia [47]. The stabilization of mast cells can decrease their response and minimize the severity in dengue, which is related to the release of VEGF and vascular permeability [48].

The ultrastructural changes that occur in the infection can be quite enlightening in relation to the processes that the cell undergoes against the pathogen. We observed degranulation of HMC-1 after 30 min of contact with the virus, but the alterations in organelles were only evident 24 h after infection. The changes caused by ZIKV were already observed in placental cells, and are consistent with those that occur in DENV, even in other cell types [25,49,50]. These changes suggest damage, mainly to mitochondria and the endoplasmic reticulum, which could impinge on the energy and protein production machinery that are necessary for viral replication. In addition, we detected the presence of virus-like particles, with the size expected for ZIKV particles, ~ 30 nm, which confirms the permittivity and ability of mast cells to replicate the virus. These observations, together with the characteristics of mast cells as an immune system component, would suggest that they would be capable of circulating throughout an affected organism, or being resident in the tissue could be responsible for cell-to-cell infection that could underlie vertical transmission.

## 5. Conclusions

Our data serves as evidence that mast cells are permissive to ZIKV infection, since a non-structural protein, NS1, was detected 24 h post infection. ZIKV can induce degranulation on its first contact and can produce cytokines and VEGF both short term and over a few hours of infection. This response of mast cells can facilitate the installation of a pro-inflammatory environment in the sites where these cells are found, such as in mucous membranes like the placenta. In addition, the fact that they can support the replication of the virus in the human placenta suggests that this type of cell may contribute to vertical transmission. Further studies are needed to fully elucidate the role of mast cells in ZIKV infection.

## Figures and Tables

**Figure 1 cells-09-00975-f001:**
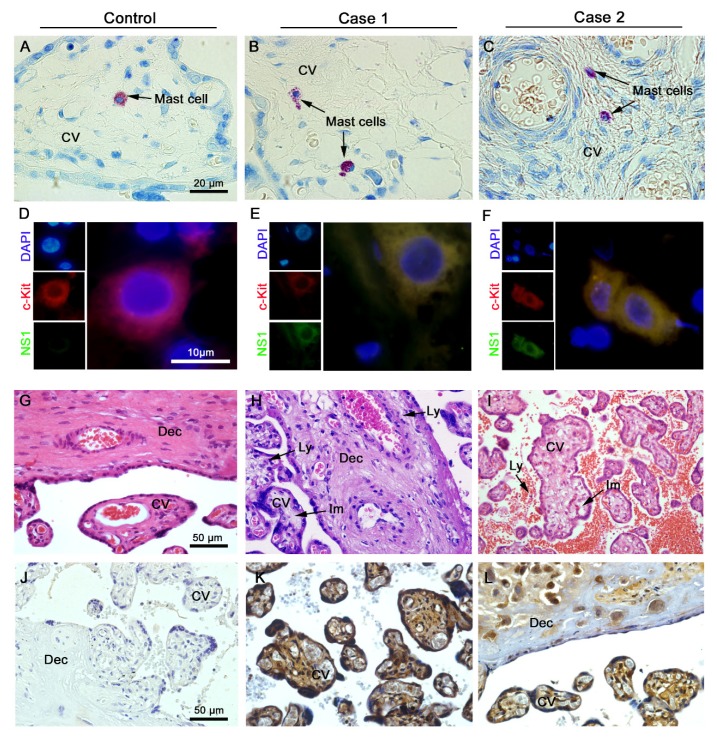
Detection of ZIKV infected mast cells in placental tissue from seropositve mothers. Placentae were collected from mothers infected or not with ZIKV immediately after childbirth and preserved in formadehyde. (**A**–**C**) Brightfield images of sections stained with Toluidine Blue showing metacromatic granules (purple, arrows) in mast calls. (**D**–**F**) Immunofluorescent images of DAPI (nuclei; blue), c-Kit (mast cell marker; red) and NS1 (ZIKV marker; green) showing ZIKV infected mast cells with both red and green fluorscence. No NS1 antigen was observed in any sections from the control placenta. The histopathological analysis of the H&E stained placentae showed normal aspects in decidua and chorionic villi within the control placenta (**G**), whereas infected placentae showed areas with lymphocytic infiltrates and immature chorionic villi (**H**,**I**). Detection of ZIKV NS1 protein by immunohistochemisty did not identify any positive cells in control placentae (**J**). Numerous cells positive for NS1 were detected in placentae from infected mothers, in both maternal and fetal portions (**K**–**L**). CV, chorionic villi; Dec, decidua; Im, immature chorionic villi; Ly, lymphocytes.

**Figure 2 cells-09-00975-f002:**
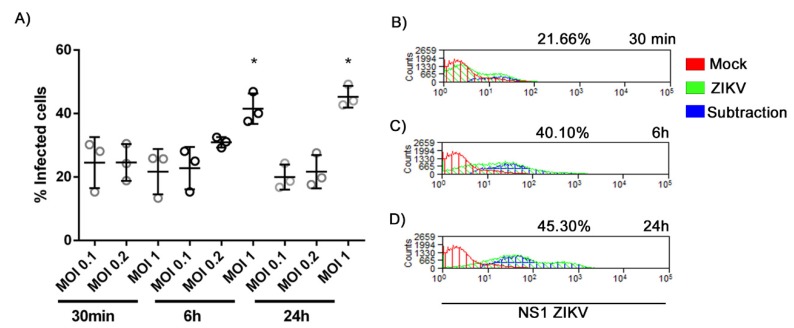
Percentage of HMC-1 cells infected with different MOIs of ZIKV. HMC-1 cells were incubated with ZIKV at MOIs of 0.1, 0.2 or 1 for 30 min and prepared for flow cytometry, or 1 h followed by 6 h or 24 h incubation before analysis. Cells were permeabilized, fixed and stained with the mouse monoclonal IgG antibody against ZIKV non-structural protein NS1 followed by incubation with the Alexa 488-conjugated anti-mouse. Panel (**A**) presents the individual percentages of HMC-1 cells expressing the NS1 protein under the different conditions from three independent experiments. Averaged histograms from the experiments with an MOI of 1 are shown in (**B**) 30 min, (**C**) 1 h with 6 h and (**D**) 1 h with 24 h infection. For negative control, cells were incubated with mock viral stocks. * Statistically significant differences between groups (same time of infection) assessed by a Mann–Whitney test (*p* < 0.05).

**Figure 3 cells-09-00975-f003:**
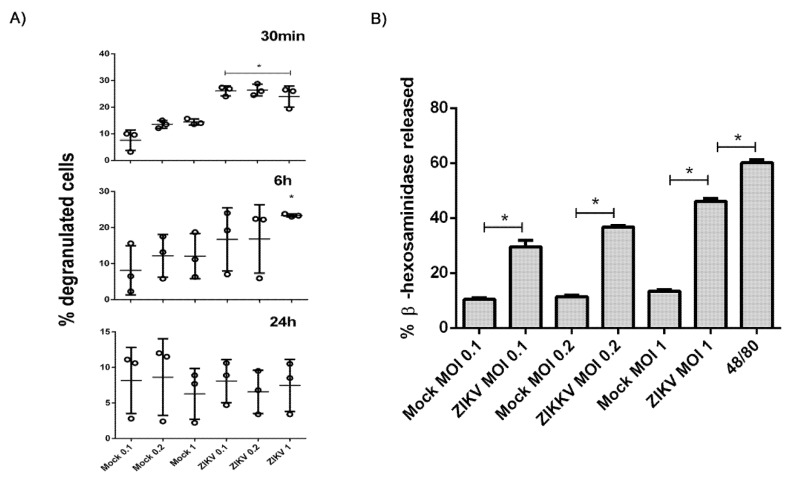
Kinetics of mast cell degranulation after interaction with ZIKV. (**A**) Percentage of degranulated cells after incubation with different MOIs of ZIKV in 30 min, 6 h and 24 h by flow cytometry. (**B**) Percentage of β-hexosaminidase release with different MOIs of ZIKV after 30 min. The synthetic compound 48/80 was used to elicite mast cell degranulation. * Statistically significant differences between groups (same time of infection) assessed by a Mann–Whitney test (*p* < 0.05). Data represent the mean of duplicate values for each sample, in three independent experiments.

**Figure 4 cells-09-00975-f004:**
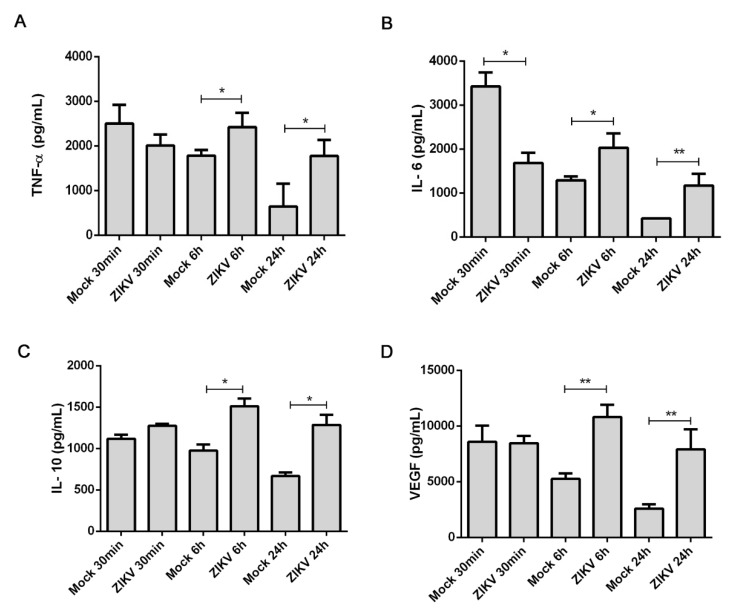
Cytokine and VEGF release by HMC-1 cells in response to ZIKV interactions. The supernatants of HCM-1 cells were collected after incubation with ZIKV or mock viral stocks for 30 min or following a 1 h incubation with an additional 6 or 24 h incubation. Commercial ELISAs were used to measure the level of released (**A**) TNF-α, (**B**) IL-6, (**C**) IL-10 levels and (**D**) VEGF. Data represent the mean of triplicate values for each sample obtained from three independet experiments. * Statistically significant differences assessed by a Mann–Whitney test (*p* < 0.05).

**Figure 5 cells-09-00975-f005:**
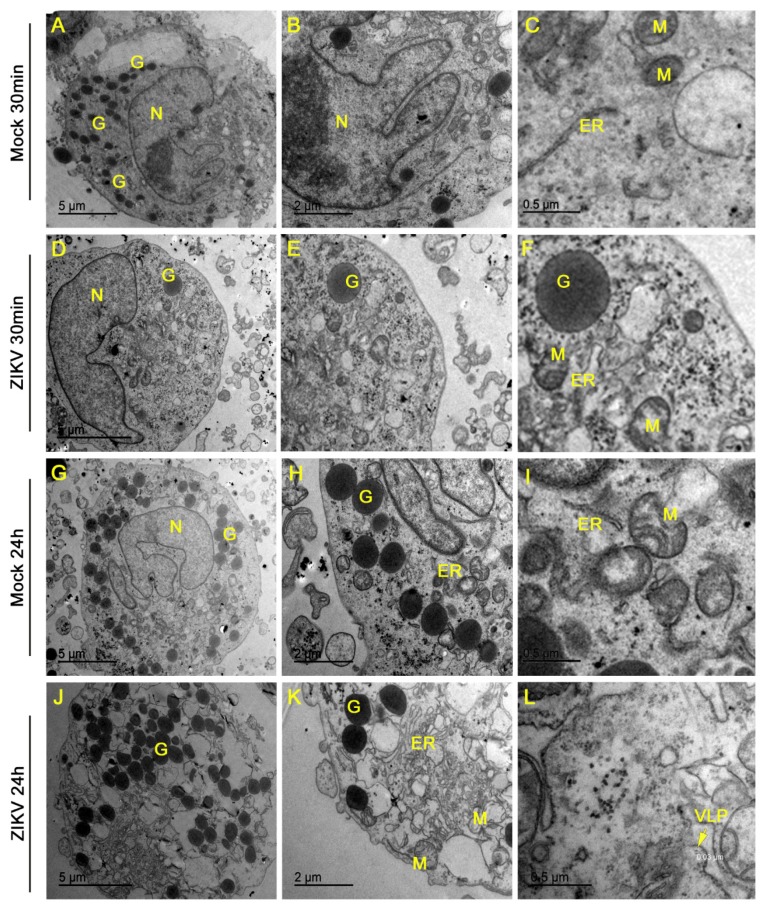
Ultrastructural changes in HMC-1 mast cells infected with ZIKV. HCM-1 cells were exposed to ZIKV or mock for 30 min or 1 h with a post 24 h incubation before processing and imaging of ultrathin sections by electron microscopy. (**A**–**C**) Control HMC-1s incubated with mock for 30 min. (**D**–**F**) An HMC-1 cell incubated with ZIKV for 30 min with decreased granules. (**G**–**I**) An HMC-1 cell incubated with mock for 24 h with a high density of granules. (**J**–**L**) HMC-1 cell infected with ZIKV for 24 h. Panel L shows a virus-like particle (VLP) with a diameter of approximately 30 nm, consistent with ZIKV. Granules (**G**), nucleus (**N**), mitochondria (**M**) and endoplasmic reticulum (ER).

**Table 1 cells-09-00975-t001:** Oligonucleotide sets to amplify ZIKV genome.

Genome Position	Region	Sequence	
835–857	M/E	sense	TTGGTCATGATACTGCTGATTGC
911–890	M/E	reverse	CCTTCCACAAAGTCCCTATTGC
860–886	M/E	probe	FAM-CGGCATACAGCATCAGGTGCATAGGAG-NFQ

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
