# Peer review of "Zika Virus Infects Human Placental Mast Cells and the HMC-1 Cell Line, and Triggers Degranulation, Cytokine Release and Ultrastructural Changes"

_cells, 2020, doi:10.3390/cells9040975_

Round 1

Reviewer 1 Report

Rabelo et al., the manuscript describes the ZIKV infection of mast cells which is interesting to some extent.

Major comments:

Line number 358, authors explained about the Vero cell has a rich secretion of proteins for the first 30 minutes whether incubated with mock or Zika virus, which is difficult to understand, the author can provide a different or better explanation (Figure 4, specifically 4B with 30 minutes of mock and ZIKV in IL-6 production).  Because the reviewer doesn't see much difference between Figure 5H and K since mast cells are responding to both mock and Zika virus by producing a degranulation effect, how the author would explain this. Above all with Figures 5A and D, at 30 minutes after exposure still, we do see more degranulation in Figure 5A.

What is the control experiment that has been performed in lieu of ZIKV, such as LPS or other toxicants? That would help better understand the degranulation effect of ZIKV.

Minor comments:

Maintain uniformity in acronyms and supplier’s address.

Author Response

Dear reviewers and editor,

We would like to thank you for all valuable comments on our study. They will certainly improve the quality of our final work. We have incorporated new data and a review of the written manuscript into this version, including a thorough review of English (all changes are highlighted in gray). About the requested experiments, briefly, we can say that we performed the histopathological analyses and the marking of ZIKV in the placental tissue, the positive control of the degranulation experiment and the ELISA for IL-10 detection. We present below a point by point response to all points raised.

Unfortunately, we can no longer go to the laboratory indefinitely, since we are in a state of emergency in Rio de Janeiro, because of the COVID-19, all people trying to cooperate with the isolation. Finally, we would like to thank the reviewers for their suggestions. We now hope that the manuscript will be appropriate for publication.

Sincerely yours,

Comment: Rabelo et al., the manuscript describes the ZIKV infection of mast cells which is interesting to some extent.

Major comments:

Line number 358, authors explained about the Vero cell has a rich secretion of proteins for the first 30 minutes whether incubated with mock or Zika virus, which is difficult to understand, the author can provide a different or better explanation (Figure 4, specifically 4B with 30 minutes of mock and ZIKV in IL-6 production). 

In fact, we quote that the mock, which has the supernatant of Vero cells, is very rich in various proteins and this may have led to the secretion of Il-6. This is not caused in these 30 minutes, the control already had these proteins. We tried to make that clearer in the discussion (lines 460-461).

Because the reviewer doesn't see much difference between Figure 5H and K since mast cells are responding to both mock and Zika virus by producing a degranulation effect, how the author would explain this. Above all with Figures 5A and D, at 30 minutes after exposure still, we do see more degranulation in Figure 5A.

Figures 5 H and 5 K correspond to the time of 24h, in which there is no degranulation of mast cells and in which these cells have already recovered from degranulation and synthesized new granules (JAMUR. Op. cit., pp. 387-388).

About figures 5A and D, there are many granules in figure 5A, approximately 30. On figure D, only 1 granule. We improved the signaling of the acronyms in the images, so that it is easier to visualize.

What is the control experiment that has been performed in lieu of ZIKV, such as LPS or other toxicants? That would help better understand the degranulation effect of ZIKV.

We use compound 48/80 as a positive control of degranulation. The data were added to figure 3B and in the text (lines 167, 313, 330 and 450).

Minor comments:

Maintain uniformity in acronyms and supplier’s address.Comments and Suggestions for Authors

We standardize acronyms and supplier addresses.

Reviewer 2 Report

The work by Rabelo et al. describe preliminary studies on contribution of mast cells during Zika virus (ZIKV) infection. Previously, a number of studies have described interaction of mast cells with different viruses including respiratory syncytial virus, rhinovirus, reovirus, dengue virus (DENV), human immunodeficiency virus and influenza, where these interactions can promote immune responses to infection but also can contribute to tissue damage. The DENV and mast cells interactions have been very well characterized in literature, so its not surprising that authors identify ZIKV presence in mast cells as ZIKV has many similarities to DENV. The presented work is very preliminary and requires more detail analysis to characterize the ZIKV infection in mast cells found in placenta and overall effect on immune and inflammatory responses.

The work lacks relevant experiments to lay foundation of the work and data needs to refined to put forward a convincing case of mast cells infection by ZIKV and effect on cytokines.

  1. In Figure 1, authors show presence of mast cells in infected patient placenta but no control (uninfected) was included. Presence of mast cells in placenta cells is expected (previously shown by Purcell and Hanahoe et al, Agents Actions 1991, Needham et al, J inflammatory Research 2016) but to assess what happens upon infected, a non infected control is warranted. Authors mention on line 84 that, ”a samples of term placenta from healthy donor was included” but data is not shown.
  2. Having access to clinical sample, one can utilize those samples to answer lot of questions, for example why authors did not show presence of ZIKV antigen using immunohistochemistry assay, it would also be important to show Histopathological analysis of the placenta of control vs infected samples. Similarly, immunofluorescence, could be performed to show localization of ZIKV antigen with mast cells in human placenta.
  3. The experiments in HMC-1 cell line are not fully characterized. Again Figure 2 lacks the mock control. At 30min time point authors don’t see any difference in infected cells but show degranulation (figure 3), how do authors explain this discrepancy?
  4. Its very surprising in the Fig2 authors don’t see a difference between infected cells at MOI 1 6 and 24h. Why authors did use another assay to show kinetics of virus infection (virus titers- plaque assay, immunostaining).
  5. The cytokine profile is not convincing. What is going on with 30 min IL6 levels? There is slight increase in cytokines upon infection at 6h but its hard to conclude from this data as mock samples show decrease over time. Has authors consider looking at other cytokines such as IL1b, IL10?
  6. The methods are not adequately described, ZIKV infection in cell lines, how virus was isolated, what primer used for detection.
  7. Fig 8 needs more clarification. It hard to make any conclusions about more or less granules

Author Response

Dear reviewers and editor,

We would like to thank you for all valuable comments on our study. They will certainly improve the quality of our final work. We have incorporated new data and a review of the written manuscript into this version, including a thorough review of English (all changes are highlighted in gray). About the requested experiments, briefly, we can say that we performed the histopathological analyses and the marking of ZIKV in the placental tissue, the positive control of the degranulation experiment and the ELISA for IL-10 detection. We present below a point by point response to all points raised.

Unfortunately, we can no longer go to the laboratory indefinitely, since we are in a state of emergency in Rio de Janeiro, because of the COVID-19, all people trying to cooperate with the isolation. Finally, we would like to thank the reviewers for their suggestions. We now hope that the manuscript will be appropriate for publication.

Sincerely yours,

The work by Rabelo et al. describe preliminary studies on contribution of mast cells during Zika virus (ZIKV) infection. Previously, a number of studies have described interaction of mast cells with different viruses including respiratory syncytial virus, rhinovirus, reovirus, dengue virus (DENV), human immunodeficiency virus and influenza, where these interactions can promote immune responses to infection but also can contribute to tissue damage. The DENV and mast cells interactions have been very well characterized in literature, so its not surprising that authors identify ZIKV presence in mast cells as ZIKV has many similarities to DENV. The presented work is very preliminary and requires more detail analysis to characterize the ZIKV infection in mast cells found in placenta and overall effect on immune and inflammatory responses.

The work lacks relevant experiments to lay foundation of the work and data needs to refined to put forward a convincing case of mast cells infection by ZIKV and effect on cytokines.

  1. In Figure 1, authors show presence of mast cells in infected patient placenta but no control (uninfected) was included. Presence of mast cells in placenta cells is expected (previously shown by Purcell and Hanahoe et al, Agents Actions 1991, Needham et al, J inflammatory Research 2016) but to assess what happens upon infected, a non infected control is warranted. Authors mention on line 84 that, ”a samples of term placenta from healthy donor was included” but data is not shown.

The photomicrograph of the mast cells in the control patient tissue was added to the work.

       2.Having access to clinical sample, one can utilize those samples to answer lot of questions, for example why authors did not show presence of ZIKV antigen using immunohistochemistry assay, it would also be important to show Histopathological analysis of the placenta of control vs infected samples. Similarly, immunofluorescence, could be performed to show localization of ZIKV antigen with mast cells in human placenta.

Based on the reviewer's suggestions, we added into Figure 1, a histopathological description and detection of the ZIKV NS1 viral antigen in the placentas of the patients cited, with their respective images.

3. The experiments in HMC-1 cell line are not fully characterized. Again Figure 2 lacks the mock control. At 30min time point authors don’t see any difference in infected cells but show degranulation (figure 3), how do authors explain this discrepancy?

Figure 2 consists of a graph made with the subtraction values obtained by flow cytometry, of three independent experiments, all with control (mock) and ZIKV. That is, the control was done (figure 2a). To make this even clear, we added the subtraction histograms, in order to exemplify how we got the values plotted in the first graph. In the histograms, it is possible to observe the intensity of fluorescence of the controls (red mock-) and infected cells (green) (figure 2b).

In relation to degranulation with 30 min, this response occurs because mast cells release their granules immediately after a stimulus, even if it is only a adsorption (to cell receptors), and not necessarily after infection. The virus takes 15 minutes to adsorb on the cell membrane and entry in the cytoplasm (van der Schaar et al., 2008), so there was enough time for degranulation. We try to make this clearer also in the discussion of manuscript (lines 447-451).

4. Its very surprising in the Fig2 authors don’t see a difference between infected cells at MOI 1 6 and 24h. Why authors did use another assay to show kinetics of virus infection (virus titers- plaque assay, immunostaining).

Yes, as studies show that the virus can enter the cell after 15 minutes (van der Schaar et al., 2008), it is normal that the number of infected cells does not vary so much between 6h and 24h. However, after 24h the cytopathic effects are more visible, so we evaluate the ultrastructure of the cells after 24h. As these cells are not adherent, performing the other experiments, such as immunostaings is really difficult. The other techniques cited, we do not have available at the moment, mainly because of our limitation with the funds in recent times in Brazil. 

5. The cytokine profile is not convincing. What is going on with 30 min IL6 levels? There is slight increase in cytokines upon infection at 6h but its hard to conclude from this data as mock samples show decrease over time. Has authors consider looking at other cytokines such as IL1b, IL10?

About the IL-6 elevated in the 30 min, we explained in the discussion, that these cytokines are stored in mast cells and can be released by external stimuli, including the proteins present in the supernatant of Vero (mock) cells (lines- 457-461). Over time, their expression increases after infection, leading to the belief that the Zika virus is the necessary stimulus to elevate the expression of this pro-inflammatory cytokine, as well as TNF-a. As requested, we carry out the experiment for the dosage of il-10 in order to make the cytokine results more consistent (Figure 4C).

6. The methods are not adequately described, ZIKV infection in cell lines, how virus was isolated, what primer used for detection.

We improved the description of the methods, we added the origin of the sample, which was from the patient's serum and the primers used.

7. Fig 8 needs more clarification. It hard to make any conclusions about more or less granules

About the transmission electron microscopy image, we use the 300 dpis definition; we expanded the brightness and contrast to improve the visualization. We have also increased the abbreviations that mark the granules to facilitate identification.

Reviewer 3 Report

The research was aimed at the development of permissive cell line for growth of the Zika virus clinical isolates in vitro. However,  the flow cytometry data revealed that less than half cells contained the Zika virus NS1 protein. Quantitation of the Zika virus replication rate is missing. So the question about permissive cell line(s) remains open.

Introduction.

According to available epidemiological reports (https://en.wikipedia.org/wiki/Zika_virus) the Zika virus is not so widely spread "in numerous tropical and subtropical countries in world". Indeed, Since the 1950s, it has been known to occur within a narrow equatorial belt from Africa to Asia. From 2007 to 2016, the virus spread eastward, across the Pacific Ocean to the Americas, leading to the 2015–2016 Zika virus epidemic.

Lne 53. The statement "Mast cells are resident and abundant in tissues such as skin, endometrium and placenta, as the main cell involved in immunologic reactions" is rather dubious. 1) The mast cells are more widely spread in contact sites with environment including respiratory tract and colon. 2) The mast cells are not "the main cells  involved in immunologic reactions. No doubt, they do invloved.

 Lines 71-72. The goal of the research does not correspond to results and conclusion.

Materials and methods.

The informed consent of both patients is not mentioned.

Line 102. "2.3 Immunofluorescence Assay"

Is it confocal fluorescent microscopy? Is NS1 protein inside the mast cells or on their surface membrane? Why the HMC-1 (both control and the Zika virus - infected) are not shown by using fluorescent microscopy?

Line 135. How multiplicity of infection was measured? What units? Pfu per cell? Or genome-equivalents per cell?

Paraformaldehyde is polymer; 4% formaldehyde (the monomer after depolymerization of paraform) is commonly used for cell fixation.

Results.

Flaviivirus infection is  relatively slow. Usually, the visible cytopathic effect appear in 24-48 hours postinfection. What happened with the Zika virus infected HMC-1 cells later, after 1 day?

What is estimated replication rate? The comparison of the Zika virus amounts before and after 1-2 days is desirable.

Innate immunity including all cytokines and chemokines is a part of protection systems. Degranulation of the mast cells may take place in response of different stress (starvation, heat and cold shock, cellular uptake of nanoparticles and so on).

Conclusion

looks like general directions and does not rely on the strict experimental evidences and, therefore, needs further revision. The described "evidence" is not enough to conclude "that mast cells are permissive to ZIKV infection".

Author Response

Dear reviewers and editor,

We would like to thank you for all valuable comments on our study. They will certainly improve the quality of our final work. We have incorporated new data and a review of the written manuscript into this version, including a thorough review of English (all changes are highlighted in gray). About the requested experiments, briefly, we can say that we performed the histopathological analyses and the marking of ZIKV in the placental tissue, the positive control of the degranulation experiment and the ELISA for IL-10 detection. We present below a point by point response to all points raised.

Unfortunately, we can no longer go to the laboratory indefinitely, since we are in a state of emergency in Rio de Janeiro, because of the COVID-19, all people trying to cooperate with the isolation. Finally, we would like to thank the reviewers for their suggestions. We now hope that the manuscript will be appropriate for publication.

Sincerely yours,

The research was aimed at the development of permissive cell line for growth of the Zika virus clinical isolates in vitro. However, the flow cytometry data revealed that less than half cells contained the Zika virus NS1 protein. Quantitation of the Zika virus replication rate is missing. So the question about permissive cell line(s) remains open.

The infection rate of 45% of cells is considered high and sufficient for the analysis, other studies with flavivirus infections such as dengue fever followed with the study even with much lower rates (Gutiérrez-Barbosa et al., 2020), or just by amplifying the viral RNA, which not shows the percentage of cells (Hermmanns et al., 2019 and Londono-Renteria et. al., 2017). In addition, the most important thing for us was to use a MOI with an acceptable amount of virus, without extrapolating the mimicking with a much higher amount of virus than cells.

Besides, when we determine the cell infected number by flow cytometry, we not only evaluate the susceptibility (ability of the virus to entry in the cell), but also its permissiveness (replication), because we detect the NS1 protein, a non structural protein that is only produced in viral replication. That is, the replication rate is not missing, both information were obtained in the same experiment. We add more information in the manuscript, to make it clearer (lines 271 and 435).

Introduction.

According to available epidemiological reports (https://en.wikipedia.org/wiki/Zika_virus) the Zika virus is not so widely spread "in numerous tropical and subtropical countries in world". Indeed, Since the 1950s, it has been known to occur within a narrow equatorial belt from Africa to Asia. From 2007 to 2016, the virus spread eastward, across the Pacific Ocean to the Americas, leading to the 2015–2016 Zika virus epidemic.

We changed the information in the introduction in order to improve the description of the places where the virus circulates (lines 39-40).

Lne 53. The statement "Mast cells are resident and abundant in tissues such as skin, endometrium and placenta, as the main cell involved in immunologic reactions" is rather dubious. 1) The mast cells are more widely spread in contact sites with environment including respiratory tract and colon. 2) The mast cells are not "the main cells  involved in immunologic reactions. No doubt, they do involved.

Information that mast cells are resident in the skin, endometrium and placenta was taken from peer-reviewed scientific articles, cited in the manuscript. We modified the manuscript and wrote that mast cells are involved in the immune response, removing determining words such as "main" and "abundant" (lines 53-54).

 Lines 71-72. The goal of the research does not correspond to results and conclusion.

Regarding the objectives, the detection of replication occurred through the detection of NS1 protein. 

Materials and methods.

The informed consent of both patients is not mentioned.

We have already added in the text that the patients were in agreement with the study and gave the appropriate consent (lines 98-100).

Line 102. "2.3 Immunofluorescence Assay"

Is it confocal fluorescent microscopy? Is NS1 protein inside the mast cells or on their surface membrane? Why the HMC-1 (both control and the Zika virus - infected) are not shown by using fluorescent microscopy?

No. As described, we performed the fluorescence microscope experiment. The NS1 protein of flavivirus is found in the cytoplasm of cells and associated with its surface, in the plasma membrane (Young et al. 2000).

Line 135. How multiplicity of infection was measured? What units? Pfu per cell? Or genome-equivalents per cell?

This information was added to the methodology, the MOI was determined through the amount of RNA measured by RT-PCR (line 130).

Paraformaldehyde is polymer; 4% formaldehyde (the monomer after depolymerization of paraform) is commonly used for cell fixation.

This information was changed in the methodology (line 148).

Results.

Flaviivirus infection is relatively slow. Usually, the visible cytopathic effect appear in 24-48 hours postinfection. What happened with the Zika virus infected HMC-1 cells later, after 1 day?

We present in our work the cytopathic effects that the virus can cause in the HMC-1 cells after 1 day of infection, with changes in the mitochondria and endoplasmic reticulum, visualized in the ultrastructure. We also evaluated the expression of cytokines, however, we did not perform tests with 48 hours of infection, because we had it in the experiments, 3 kinetic times and one of the main responses in mast cells is degranulation, which happens in immediate contact with the virus, therefore the focus in the initial times.

What is estimated replication rate? The comparison of the Zika virus amounts before and after 1-2 days is desirable.

Innate immunity including all cytokines and chemokines is a part of protection systems. Degranulation of the mast cells may take place in response of different stress (starvation, heat and cold shock, cellular uptake of nanoparticles and so on).

As explained earlier, we did this by cytometry, with the detection of NS1 protein. Unfortunately, we have no way to perform a real-time PCR experiment, for example, due to lack of resources and time. In Brazil we are going through a critical period of funds cut and now, exactly, we cannot go to the laboratory through the state of emergency in Rio de Janeiro with COVID 19.

Conclusion

looks like general directions and does not rely on the strict experimental evidences and, therefore, needs further revision. The described "evidence" is not enough to conclude "that mast cells are permissive to ZIKV infection".

About our conclusion, we emphasize that the detection of the NS1 protein is one of the ZIKV proteins involved in replication, a proof of permissiveness, since it is not part of the virion, but is produced only during the ZIKV replication cycle.

Round 2

Reviewer 2 Report

Authors have addressed all the reviewer's comments. Authors are recommended to go over manuscript edit minor typos and spell checks.

Author Response

.